# Sensitive Electrochemical Non-Enzymatic Detection of Glucose Based on Wireless Data Transmission

**DOI:** 10.3390/s22072787

**Published:** 2022-04-05

**Authors:** Young-Joon Kim, Somasekhar R. Chinnadayyala, Hien T. Ngoc Le, Sungbo Cho

**Affiliations:** 1Department of Electronic Engineering, Gachon University, 1342 Seongnam-daero, Seongnam 13120, Korea; youngkim@gachon.ac.kr; 2Sensors and Aerosols Laboratory, Department of Mechanical Engineering, Ulsan National Institute of Science and Technology (UNIST), Ulsan 44919, Korea; ssreddy@unist.ac.kr; 3Gachon Advanced Institute for Health Science & Technology, Gachon University, 155 Gaetbeol-ro, Incheon 21999, Korea

**Keywords:** microneedle electrode array, continuous glucose monitoring, wireless data transmission, chronoamperometry, platinum black, glucose sensor

## Abstract

Miniaturization and wireless continuous glucose monitoring are key factors for the successful management of diabetes. Electrochemical sensors are very versatile and can be easily miniaturized for wireless glucose monitoring. The authors report a microneedle-based enzyme-free electrochemical wireless sensor for painless and continuous glucose monitoring. The microneedles (MNs) fabricated consist of a 3 × 5 sharp and stainless-steel electrode array configuration. Each MN in the 3 × 5 array has 575 µm × 150 µm in height and width, respectively. A glucose-catalyzing layer, porous platinum black, was electrochemically deposited on the tips of the MNs by applying a fixed cathodic current of 2.5 mA cm^−2^ for a period of 200 s. For the non-interference glucose sensing, the platinum (Pt)-black-coated MN was carefully packaged into a biocompatible ionomer, nafion. The surface morphologies of the bare and modified MNs were studied using field-emission scanning electron microscopy (FESEM) and energy-dispersive X-ray analysis (EDX). The wireless glucose sensor displayed a broad linear range of glucose (1→30 mM), a good sensitivity and higher detection limit of 145.33 μA mM^−1^ cm^−2^ and 480 μM, respectively, with bare AuMN as a counter electrode. However, the wireless device showed an improved sensitivity and enhanced detection limit of 445.75, 165.83 μA mM^−1^ cm^−2^ and 268 μM, respectively, with the Pt-black-modified MN as a counter electrode. The sensor also exhibited a very good response time (2 s) and a limited interference effect on the detection of glucose in the presence of other electroactive oxidizing species, indicating a very fast and interference-free chronoamperometric response.

## 1. Introduction

Diabetes mellitus, instigated primarily by the lack of insulin, is a common human health problem identified by increased levels of glucose concentration in the blood [1]. For a healthy individual, the physiological levels of blood glucose range from 4.4 to 6.6 mM [2]. Nevertheless, any abnormality from the physiological range for a prolonged period of time can cause numerous severe disorders such as diabetic retinopathy, nephropathy and heart attack [3]. Tight metabolic control using frequent glucose measurements can establish a guide for the administration of insulin, but it is difficult to maintain normoglycemia without blood glucose concentrations intermittently declining or increasing to an excessive level. Therefore, high-accuracy, low-cost and continuous glucose sensors that can provide glucose information throughout the day have gained attention worldwide [4]. Furthermore, the continuous glucose measurement with high selectivity and sensitivity has gained considerable interest in various sectors such as the food industry, environmental monitoring and clinical diagnostics. Amid the different detection techniques available for the detection of varying glucose levels in the body, electrochemical detection methods have attracted significant attention due to their ability for mass production, their speed, their low cost, they are easy to commercialize, their high sensitivity, their use in point-of-care testing and the facile glucose electro-oxidation on the electrode surface [5,6,7]. Along with the glucose sensors, various wireless measurement methods are realized for continuous monitoring, but real-time validation, including an application-based smartphone display, has especially received publicity from the patients and clinical staff [8,9].

In the past two decades, enzyme-based electrochemical glucose sensing has been widely studied, and the most viable glucose sensing test strips available in the commercial market are based on glucose oxidase and glucose dehydrogenase enzymes immobilized onto a suitable matrix [10,11]. Nevertheless, the sensitive enzymatic glucose sensors are associated with several drawbacks such as instability (pH, temperature and humidity) due to the denaturation of the enzyme, oxygen limitations and complex immobilization procedures [12]. Owing to the fast growth of the world population, ~422 million people in 2020 suffered from diabetes, and the number is expected to reach 592 million by 2035 [13]. Therefore, the development of an accurate, sensitive and low-cost electrochemical glucose sensor is of great need. In contrast, the enzyme-free detection of glucose on various noble metals (platinum, gold, silver and alloy metals) has been studied extensively. Electrodes modified with nonporous noble metals show enhanced electro-catalytic activities due to higher conductivity, increased electron transfer rate and large surface area [14,15,16,17]. Non-enzymatic glucose sensors have attracted significant interests to exploit their direct electrocatalytic detection and approve cost-effective fabrication and high stability. The following are some examples: the development of an efficient, sensitive, and non-enzymatic electrochemical sensor for glucose detection in an alkaline media based on Ag@TiO_2_@ Metal–Organic Framework (ZIF-67) nanocomposite with high sensitivity of 0.788 μA μM^−1^ cm^−2^ [12]; the construction of a non-enzymatic amperometric glucose sensor working at very low applied potential of 0.2 V based on a nano-TiO_2_-coated aluminum tip electrode with a lower detection limit of 0.55 mM of glucose [18]; the development of a highly efficient non-enzymatic glucose sensor with high sensitivity and low limit of detection (2961.7 μA mM^−1^ cm^−2^ and 0.40 μM, respectively) based on CuO modified vertically grown ZnO nanorods on electrode [19]; and the construction of a non-enzymatic glucose sensor based on a CoNi_2_Se_4_/rGO nanocomposite with ultrahigh sensitivity (18.89 mA mM^−1^ cm^−2^) at low working potential of +0.35 V [20].

Several reports on microneedles (MNs) have been published in the past decade due to their attractive features such as being minimally invasive for painless sampling and micron-size dimensions limiting their penetration to the intradermal space, thanks to the MEMS fabrication technology [21,22]. To date, MNs have been used for transdermal drug delivery [23], insulin injection [24] and for sensing various analytes of clinical interest lactate [25], glucose [22], L-DOPA [26], glutamate [27] and cholesterol [28]. Recently, minimally invasive non-enzymatic glucose monitoring sensors based on porous metals have gained considerable research interest as alternatives for enzyme-based glucose sensors due to their direct electro-oxidation of glucose on the surface of MNs. 

For a highly selective and interference-free detection of glucose, usually, a very thin coating of a membrane based on size or charge exclusion is used on the surface of the modified electrodes (e.g., cellulose acetate, polypyrrole, nafion, poly(vinylpyridine)) [29]. In the present study, a charge exclusion membrane, nafion, was selected to coat the MNs modified with Pt-black due to its biocompatibility, electrical conductivity, good chemical stability, strong exclusion to the negatively charged proteins and common interfering redox active agents, easy availability and simple handling [30].

Continuous glucose monitoring is mostly achieved by a wearable wireless instrument. For the patient’s comfort and ease of use, the device must be small and low power. In addition, the minimum detectable current should be small enough for an accurate measurement. In this study, a dedicated electrochemical front end (EFE) and a Bluetooth low energy (BLE) integrated microprocessor unit (MCU) were selected for its low power operation and small footprint. The amperometric current is measured by the device and wirelessly transmitted to an external smartphone or PC for real-time analysis. A dedicated user interface displays the measurement result. 

Herein, we investigated an enzyme-free porous noble metal (Pt-black)-coated microneedle electrode array (MNEA)-based sensor for sensing glucose in vitro. A miniaturized readout system is fabricated for continuous glucose monitoring. The readout system is composed of an EFE and a BLE integrated MCU, where the MCU controls the EFE by programming the digital-to-analog converter (DAC) to an appropriate electrical potential for the sensor excitation and detects the amperometric current by sampling the measured data with a 16-bit analog-to-digital converter (ADC). The measurement unit is highly integrated by utilizing chip-scale package integrated circuits (IC), achieving a system size of 30 mm × 20 mm × 5 mm, including the battery and biocompatible package. The nonporous platinum black layer was deposited electrochemically using a cathodic current density (2.5 mA cm^−2^ vs. Ag/AgCl external reference electrode) and from an electrolytic bath containing hexachloroplatinic acid, lead acetate and hydrogen chloride. The modified MNEAs were packed using anion exclusion membrane for non-interference glucose detection. The modified MNs were analyzed for surface morphology (SEM/EDS) and electrochemical glucose sensing in PBS using a custom designed readout instrument, which wirelessly transmits the measured data to a mobile phone (Samsung, Suwon, Korea, SM-G986). An Android application reconstructs a real-time amperometric plot from the received data in a user-friendly interface for a direct analysis.

## 2. Materials and Methods

### 2.1. Reagents and Chemicals

Hexachloroplatinic(IV) acid hexahydrate (H_2_PtCl_6_·6H_2_O) (≥37.50% Pt basis), lead diacetate trihydrate (≥99%), hydrochloric acid (HCl), nafion (5 wt.%, solution), D-(+)-glucose (≥99.5%), ascorbic acid (98%), lactic acid (≥98%), dopamine hydrochloride, d-(+)-galactose (≥99%), d-(+)-mannose (≥99%), d-(–)-fructose (≥99%) and uric acid (≥99%) and urea (99–100.5%) were procured from Sigma Aldrich (St. Louis, MO, USA). Moreover, 100 mM phosphate-buffered saline (PBS, pH = 7.4) and ethanol (99%) were purchased from OCI (Seoul, South Korea). All other chemicals were of analytical grade and used as received without any further purification. 

### 2.2. Material Characterization

Scanning electron microscopy images and energy-dispersive X-ray analyses were obtained on a field emission SEM equipped with EDS spectrometer (HITACHI S-4700; Tokyo, Japan) operated at a voltage of 15 kV to discern the surface morphology and the elemental analysis of the modified MNs. 

### 2.3. Gold Microneedle (MN) Fabrication

MN patterns with specific dimensions were designed using AutoCAD (Autodesk Inc., San Rafael, CA, USA) (version 2020). A jet of wet chemical etchant (ferric chloride, FeCl_3_) under a pressure of 2 kgf/cm^2^ for a time period of 60 s was used to pattern a 316 L Grade stainless-steel substrate (150 µm). A thin gold layer (50 nm) was electroplated on the patterned stainless-steel substrate, and the microneedle electrode arrays (MNEAs) fabricated were bent out of the plane (90°) using a jig. A PDMS block and parafilm were used to conceal the tip positions and contact pad of MNs to avoid the parylene coating on the respective regions. A thin layer of parylene (5 µm) was used to passivate the Au layer by using a parylene coating system (LAVIDA-110H) from Femto Science Inc. (Hwaseong, Korea). The microneedle tip positions and contact pads were unmasked from PDMS block and the parafilm, respectively, and the unpassivated MN tip positions were used to electrodeposit the Pt-black catalytic layer.

### 2.4. Preparation of Pt-Black Microneedles

A three-electrode electrochemical set-up was used to electrodeposit a porous Pt-black sensing layer on the fabricated Au-MNs (Appendix A). A cathodic deposition was conducted in an aqueous solution of 2.5% chloroplatinic acid hexahydrate, 0.05% lead diacetate trihydrate and 0.01 M of HCl by applying a fixed cathodic current density of 2.5 mA cm^−2^ vs. Ag/AgCl external reference electrode for a time period of 200 s. After electrodeposition, the MNs were washed gently with DI water.

### 2.5. Packaging of the Pt-Black Microneedles

A bio-compatible fluoropolymer, nafion, was used for packaging of the modified MNs. A dip-coating method using a 1:6% *v*/*v* nafion: ethanol mixture was used to coat the developed Pt-black MNs. Briefly, the modified needles were held at contact pads using a surgical forceps and dip-coated in 1:6% *v*/*v* nafion: ethanol mixture for 60 s and dried at 50 °C on a hot plate for 80 s. The dip-coated microneedles were subsequently air dried at room temperature (RT) (24 °C) for 12 h before carrying out the electrochemical measurements.

### 2.6. Fabrication of the Readout System 

The device in Figure 1 is a wireless, non-invasive system for real-time continuous glucose measurement. A thin layer of FR (0.5 mm) serves as the substrate of the electronic device. Soldering paste (TS391LT, Chip Quik, Ancaster, ON, Canada) was used to populate the various surface-mounted components, including the antenna (2450AT07A0100, Johanson Technology, Camarillo, CA, USA). Chip-scale packaged ICs, including EFE (AD5940, Analog Devices, Tokyo, Japan) and BLE SoC (nRF52832, Nordic Semiconductor, Trondheim, Norway) were assembled to the PCB at 280 °C using a heat gun (861DW, Quick, Jiangsu, China). The external electrode interconnections were established by using thin copper wires. The firmware was programmed and flashed to the BLE SoC by a wired connection. Once the initial upload is complete, further software change can be made wirelessly by BLE. To prevent electrical shorts, an insulation tape is placed between the PCB and the battery. A 3-V button cell battery (CR2032, Panasonic, Osaka, Japan) is mounted on the bottom of the PCB for power supply. Soft layers of PDMS (Sylgard 184, Dow Corning, Midland, MI, USA) were molded to form a robust encapsulating structure. A 10:1 mixture of Sylgard 184 base and curing agent was cured at room temperature for 24 h. Total dimensions of the device, including the electronic instrument, battery and the PDMS encapsulation layer, are measured as 30 mm × 24 mm × 5 mm.

### 2.7. Electrochemical Measurements

Electrochemical analysis of the modified MNEAs was carried out in a three-electrode setup on fabricated wireless device, consisting of a working electrode (WE), (modified MNs) a counter electrode (CE) (Bare AuMN electrode array or Pt-black-modified MNEA) and a reference electrode (RE) (AgCl MN). The amperometric measurements were recorded in 100 mM PBS (pH = 7.4, 24 °C) using a 3-electrode setup at a constant electrode applied potential (*E_app_*) of + 0.12 V vs. AgCl MNEA at RT (24 °C) (Appendix A). 

Figure 2 presents the block diagrams of the EFE and BLE SoC for wireless communication. The data processing and serial communication interface (SPI) is managed by the MCU, and a 12-bit DAC generates a constant + 0.12 V between the WE and RE. The power amplifier delivers the required charges to the CE, while the transimpedance amplifier magnifies the current signal. The dynamic range of the current signal is internally selected by the MCU through a programmable resistor (RTIA) and a programmable-gain amplifier. A current ranging from 50 pA to 3 mA can successfully be amplified and sampled by a 16-bit analog-to-digital converter (ADC). The EFE executes digital signal processing on the ADC-sampled data to filter out high frequency noise, and the data are stored in the internal SRAM at 10 SPS. The interrupt controller generates an interrupt signal when 40 data sets are saved, and this data packet is transferred to the BLE SoC via SPI communication. Wireless transmission occurs over the BLE radio to the user interface (Android smartphone, PC) where the data are displayed as an amperometric current plot in real-time. A real-time infinite impulse response (IIR) filter is implemented to reduce unnecessary glitches or abrupt changes. While the instrument is continuously measuring the glucose concentration, the average current consumption is approximately 4 mA, which ensures a battery life-time of 56 h. For an easy battery replacement and portability, a battery holder can be used with a smaller sized battery (CR1220). A reliable Bluetooth pairing is achieved over a 10-m distance.

Once the measurement data have been received by the host smartphone, the device not only displays the real-time current measurement in the sensor but also stores the measurement data in the local storage and displays the current glucose level based on the measurement (Figure 3). Upon initial Bluetooth establishment, the host device does not display any prediction for glucose concentration. It waits 60 s for a current stabilization and takes the current reading at this instance as an offset current ($i_{\mathrm{offset}}$). After this stabilization period, the prediction algorithm displays the estimated glucose concentration level by using the predetermined linear regression of the sensor response. While monitoring the glucose level, the user can actively change the parameters of the prediction algorithm through their smartphone. While the instrument is continuously measuring the glucose concentration, the average current consumption is approximately 4 mA, and a solid Bluetooth pairing is achieved over a 10-m distance.

## 3. Results and Discussion

### 3.1. Surface Characterization of MNEAs

The surface structure analyses of the bare MNEAs and Pt-black-modified MNEAs were characterized by FESEM. An exceptional needle consistency, as indicated by the AutoCAD design with a smooth surface, was observed for the bare MNs (Figure 4a,b). The bare Au-MNs shows a length and width of 575 µm × 150 μm, respectively. The SEM of modified MNs shows the continually patterned and extended branch-like structures of the nanoporous Pt-black (Figure 4c). The patterned and extended branch-like structures of the Pt-black were converted into a nanoporous morphology with a smooth surface upon coating with the nafion ionomer (Figure 4d). Well-defined nanoporous-patterned structures were formed at an optimal cathodic current density and electrodeposition time. A cathodic current density of 2.5 mA cm^−2^ for a period of 200 s is optimized for the formation of well-defined porous Pt-black and to entirely coat the MN surface with no damage to the electroplated Au layer on the MNs. Uniform and clear nanoporous structures were formed on the MN tips when the distance between the WE and the CE was larger than 2 cm. 

The EDX analysis was performed to validate the elementary components, atomic and weight ratios of the modified MNs (SEM/EDX, Figure 5), by taking the measurements over random regions (purple region in the inset). Figure 5a, shows the presence of gold as a major component in bare AuMN electrode array suggesting the uniform coating of thin gold layer on the stainless-steel substrate. Figure 5b shows the presence of platinum as major components in AuMNs/Pt-black. The EDX of AuMNs/Pt-black/Nf shows the platinum as major component and carbon, fluorine sulfur and oxygen as minor components, suggesting the deposition of nanoporous Pt-black and packaging of the Pt-black MNs with the nafion ionomer (Figure 5c). 

### 3.2. Chronoamperometry for Glucose Sensor

Figure 6a shows the constant potential *i* vs. *t* response curves of the Au/Pt-black/Nf MNEA measured with wireless non-invasive CGM system fabricated in the present study with bare AuMN as the CE for the step-wise addition of glucose to study the analytical characteristics of the Pt-black-modified MNEAs. Our previous article reported the potential window of the glucose oxidation and suggested +0.12 V vs. Ag/AgCl as an optimized potential for chronoamperometry of AuMNs/Pt-black/Nf sensors [5]. Additionally, the potential window of the glucose oxidation (+0.0 V to +0.3 V) was shown in Appendix A describing the cyclic voltammograms of the bare Au MNEA, Au/Pt-black and Au/Pt-black/Nf without or with the addition of 10 mM of glucose at a scan rate of 50 mV s^−1^ in 10 × PBS (pH = 7.4). The response curves were recorded at a fixed external potential of +0.12 V (*E_app_* = +0.12 V vs. AgCl MN) with the successive step addition of glucose over a concentration range of 1 to 30 mM. The constant potential of +0.12 V vs. AgCl MN was carefully chosen to prevent the effect of interference from electro-active interfering agents present in the biological samples. The performance characteristics of the sensor measured with the system was summarized in Table 1. At the fixed potential of +0.12 V vs. AgCl MN, the current response of the MNEA sensor increased significantly and achieved the steady-state current (95%) within 2 s, suggesting a rapid chronoamperometric response. 

The modified MNs (Au/Pt-black/Nf) show a linear increase in the current response at +0.12 V vs. AgCl MN with the step-wise addition of glucose under constant stirring, and the electrode shows a dynamic range of 1–30 mM for both measurement systems. The developed sensor has a limited dynamic measurement range and cannot detect too small or too large concentrations of glucose. It was shown that the current response of the sensor reached a saturation level at the glucose concentration higher than 40 mM. Figure 6b shows the linear calibration curve of the sensor response, demonstrating a good linear range detecting from a physiological level to pathological levels of glucose. The electrode shows a detection limit of 480 μM for the wireless non-invasive CGM system with bare AuMN as the CE, respectively. The detection limit is calculated using S/N = 3, (3 × SD/slope), where SD is the standard deviation of the blank electrode response and slope is the sensitivity of the modified electrode [28]. The MNEA sensor shows a sensitivity of 145.33 μA mM^−1^ cm^−2^ for the wireless non-invasive CGM system measured with the bare AuMN as the CE, respectively. 

In order to attain a better sensitivity and detection limit in the modified electrodes, the bare Au MNEA counter electrode is replaced with a Pt-black-modified MNEA as a new counter electrode, and the amperometric current responses are measured with the developed fabricated non-invasive wireless CGM system. Since the Pt-black-modified MNEA provided a higher surface area for the counter electrode’s comparison to the bare Au MNEA, the current flow between WE (AuMN/Pt-black/Nf) and CE (AuMN/Pt-black) increased, which resulted in an enhancement in the glucose measurement sensitivity. The constant potential chronoamperometric response curve measured using the Pt-black/MNEA as a new counter electrode for the non-invasive wireless device is shown in Figure 7. The performance characteristics of the sensor measured with the system were summarized in Table 1. The modified MNEAs (Au/Pt-black/Nf) show an increase in current response at +0.12 V vs. AgCl MN with the step-wise addition of glucose under constant stirring, and the electrode shows a dynamic range of 1→30 mM for the measurement system.

The electrode shows a detection limit of 268 μM for the wireless non-invasive CGM system with the Pt-black-modified MNEA as a CE. The MNEA sensor shows a sensitivity of 445.75 and 165.83 μA mM^−1^ cm^−2^ measured with the wireless device with Pt-black-modified MNEA as a CE. The sensitivity of the modified electrode (AuMN/Pt-black/Nf) with Pt-black as a CE shows a greater value compared to the sensitivity of the modified electrode (AuMN/Pt-black/Nf) with bare Au MNEA as a CE. The results suggest that the increase in the electrode surface area upon Pt-black coating on the CE enhances the current balance between the WE and CE during the successive step addition of glucose, thus increasing the sensitivity of the developed Au/Pt-black/Nf MNEA sensor, suggesting a great performance of the fabricated wireless non-enzymatic device as compared to other glucose sensors in Table 1. It is evident from the results that the fabricated device can be used as a portable CGM device in the near future for diabetic applications. The sensitivity recorded in the current study using the wireless device (Au/Pt-black/Nf MNEA) is greater than the several non-enzymatic glucose sensors reported earlier [29,30,32,33,34,35]. The linear range reported in the present study is greater than many of the enzyme-free glucose sensors reported in the past decade [29,30,31,32,33,34,35,36,37].

### 3.3. Interference Study and Application in Interstitial Fluid (ISF)

Selectivity is one of the challenging factors for the enhanced performance of the electrochemical sensors. During real sample analysis, the co-existence of other electroactive interfering agents, such as ascorbic acid, uric acid, lactic acid, urea, acetaminophen, dopamine, mannose, fructose, galactose and sodium chloride, might affect the glucose detection. Hence, the effect of interferents on the Au/Pt-black/Nf electrode is measured at concentrations higher (10-fold) than the physiological level of the interferent using the fabricated wireless CGM device. Figure 8 shows the constant potential chronoamperometric interference study carried out on the Au/Pt-black/Nf electrode using a wireless device. The noise in the amerometric current response was observed immediately after the glucose addition. However, the current response became stable and kept their increased level. The interference results of the wireless device demonstrate that the corresponding change in oxidation current upon the addition of 5 mM glucose is 19.52 μA, which is significantly higher than those recorded for the added interferents 0.41 μA for AA (5 mM), 0.2 μA for UA (10 mM), 0.55 μA for LA (10 mM), 0.1 μA for U (10 mM), 0.37 μA for AP (5 mM), 0.15 μA for DP (10 mM), 0.15 μA for Man (10 mM) 0.1 μA for Fru (10 mM), 0.19 μA for Gal (10 mM) and 0.14 μA for NaCl. The interference results endorsed that the developed wireless system could measure the slight change in the oxidation current upon the addition of interferents, and any of the added interferents caused a negligible change in the current response. The insignificant current response measured in the presence of oxidizing electroactive interferents is due to the hydrophobic underlying electrode packaging material, nafion, which exerted a very limited access of the interferents onto the electrode surface due to the strong electrostatic repulsions excreted by the hydrophobic side chains of the polymer [38,39,40,41]. The interference-free detection of glucose measured by the fabricated wireless device shows a selective electrochemical activity of the Pt-black layer towards glucose addition.

Our previous studies showed the feasibility of the microneedle-based sensor array for the continuous monitoring of glucose in the interstitial fluid (ISF) and animal model [5,6]. Here, ISF was prepared by adding 2.5 mM CaCl_2_, 5.5 mM glucose, 10 mM HEPES (pH: 6.8), 3.5 mM KCL, 0.7 mM MgSO_4_, 123 mM NaCl, 1.5 mM NaH_2_PO_4_ and 7.4 mM saccharose. The solution pH was adjusted to 7.0 using 1 N HCl. The sensor showed a response in the ISF in a range of concentration from 1 mM to 30 mM, as seen in Figure 9, showing the feasibility of the sensor to real samples. The recovery test of the sensor showed a good recovery range (98.0~102.5%; *n* = 5) and no significant difference (*p* = 0.572) between the spiked (3.0, 5.0, 8.0, and 12.0 mM) and the measured glucose concentrations in ISF samples [6].

The repeatability and storage stability of the sensor for practical application was tested in our previous articles [6]. From the repeatability evaluation of the fabricated MNEAs, the measured intra-assay relative standard deviation was 1.64 %, demonstrating that the MNEA sensor has satisfactory repeatability. The storage stability test showed a 3.5% loss in the initial response at the end of day 16, probably due to a disturbance in the nafion packaging.

## 4. Conclusions

The present study endorsed the utilization of a microneedle, enzyme-free sensor for the continuous monitoring of glucose in vitro. The significance of this work is the utilization of microneedle-modified nanoporous Pt-black nanostructures as a glucose catalytic layer and the nafion ionomer as a biocompatible coating polymer for wireless and non-interference glucose detection. The major outcomes of the study include the (a) miniaturization of the device, (b) wireless data transmission to an external smartphone through the integration of an electrochemical front end (EFE) and (c) a Bluetooth low-energy (BLE)-integrated microprocessor unit (MCU) for its low power operation and small footprint. The Pt-black modification of the counter electrode (bare AuMN) increased the electroactive surface area of the counter electrode, thus improving the analytical characteristics of the developed wireless sensor (sensitivity and detection limit). The electrode array displayed a good selectivity in the presence of 10-fold higher concentration of electroactive oxidizing species due to the electrorepulsion of the packaging polymer nafion. The developed non-enzymatic AuMN/Pt-black/Nf array sensors are promising devices for the continuous monitoring of glucose in individuals with diabetes. Future studies include the minimally invasive and wireless glucose sensing in animal models. 

## Figures and Tables

**Figure 1 sensors-22-02787-f001:**
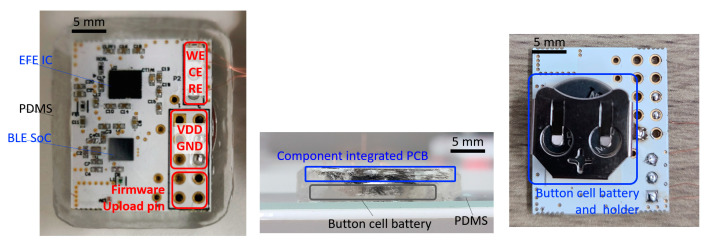
Image of the wireless glucose monitoring device. Top, side view of the PDMS packaged device and bottom view of the device with a battery holder.

**Figure 2 sensors-22-02787-f002:**
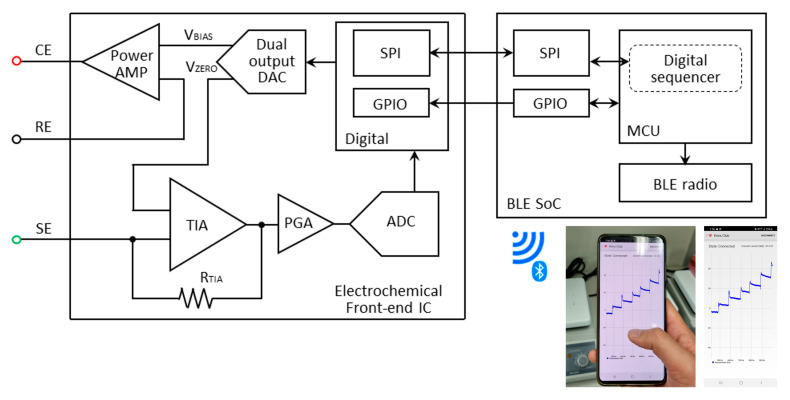
Block diagram of the wireless glucose monitoring device.

**Figure 3 sensors-22-02787-f003:**
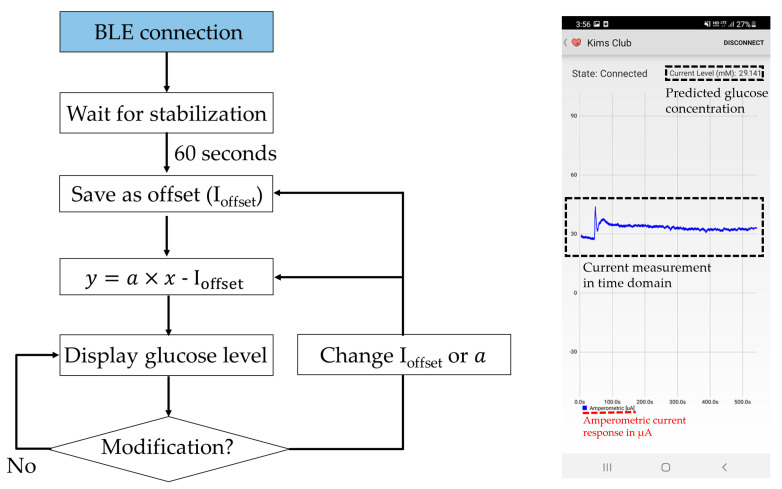
The glucose prediction algorithm and the user interface. I_offset_, a and y denote the stabilized offset current, linear current-glucose sensitivity and glucose concentration, respectively.

**Figure 4 sensors-22-02787-f004:**
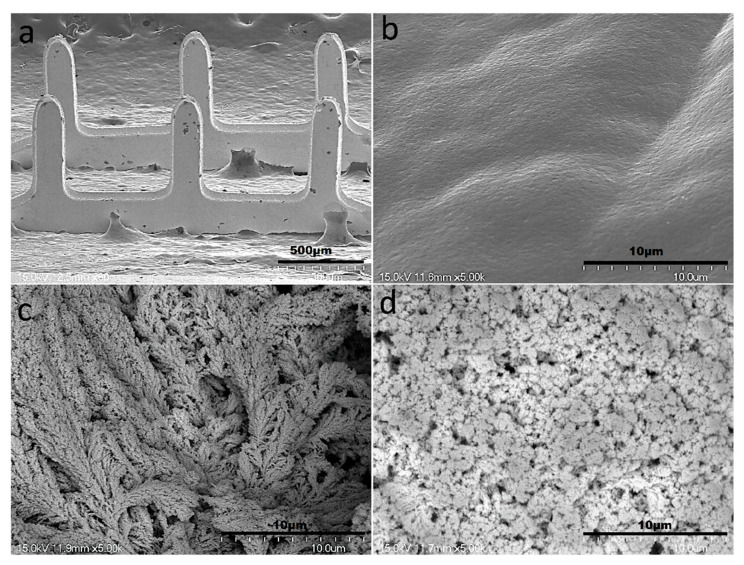
SEM images of the bare AuMN electrode arrays displaying an exceptional needle consistency (**a**); Smooth surface morphology of the bare AuMN (**b**); turning into extended branch-like structures upon Pt-black electrodeposition (**c**); and nanoporous morphology with smooth surface after packaging with the nafion polymer (**d**).

**Figure 5 sensors-22-02787-f005:**
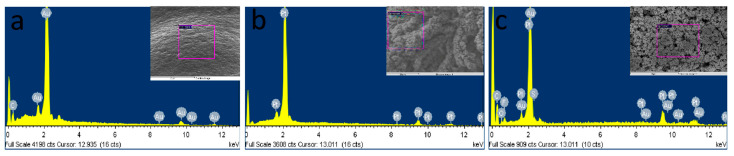
SEM/EDX characterization of the bare AuMN (**a**), AuMN/Pt-black (**b**) and AuMN/Pt-black/Nf electrodes (**c**).

**Figure 6 sensors-22-02787-f006:**
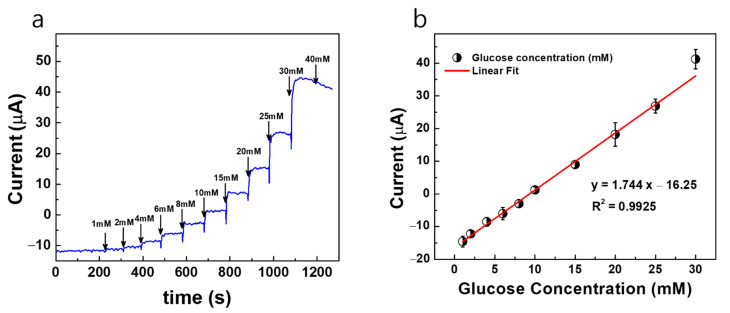
(**a**) Constant potential amperometric current response curves (*i* vs. *t*) for AuMN/Pt-black/Nf electrode array measured using the fabricated wireless non-invasive CGM system at a fixed external applied potential of +0.12 V vs. AgCl MN and bare AuMN as counter electrode in a background electrolyte of 100 mM PBS with successive step-wise addition of glucose from 1 mM→40 mM; (**b**) the linear fitting curves for the current responses measured with the correlation co-efficient values (R^2^) and linear regression equations. The measured data are presented as the average of three independent experiments (*n* = 3) with the range represented as standard error.

**Figure 7 sensors-22-02787-f007:**
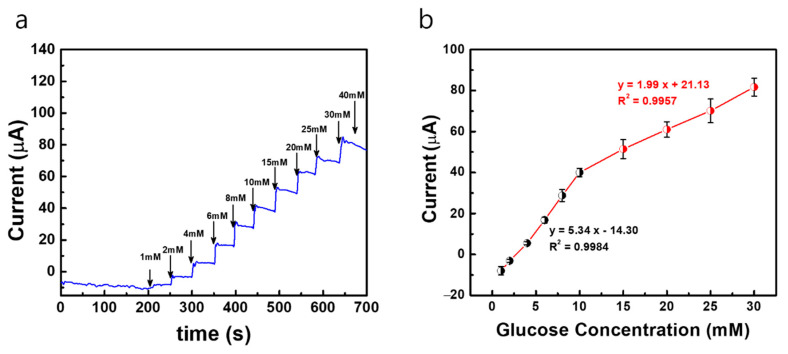
(**a**) Constant potential amperometric current response curves (*i* vs. *t*) of AuMN/Pt-black/Nf electrode array measured using the fabricated wireless non-invasive CGM system at a fixed external applied potential of +0.12 V vs. AgCl MN and Pt-black-deposited MN as counter electrode in a background electrolyte of 100 mM PBS with successive step-wise addition of glucose from 1 mM→40 mM; (**b**) the linear fitting curves for the current responses measured with the correlation co-efficient values (R^2^) and linear regression equations. The measured data are presented as the average of three independent experiments (*n* = 3) with the range represented as standard error.

**Figure 8 sensors-22-02787-f008:**
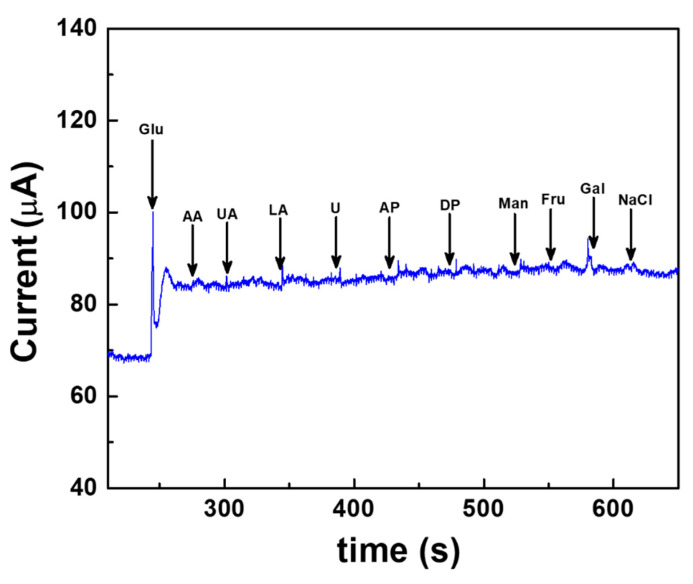
Interference-free measurement of glucose measured using the fabricated wireless device at a fixed external potential of +0.12 V vs. AgCl electrode carried out in a background electrolyte of 100 mM PBS (pH 7.4). The common interfering agents present in real samples such as ascorbic acid (AA), uric acid (UA), lactic acid (LA), urea (U), acetaminophen (AP), dopamine (DP), mannose (Man), fructose (Fru), galactose (Gal) and sodium chloride (NaCl) are spiked at a regular interval of 50 s and at concentrations greater than the physiological levels.

**Figure 9 sensors-22-02787-f009:**
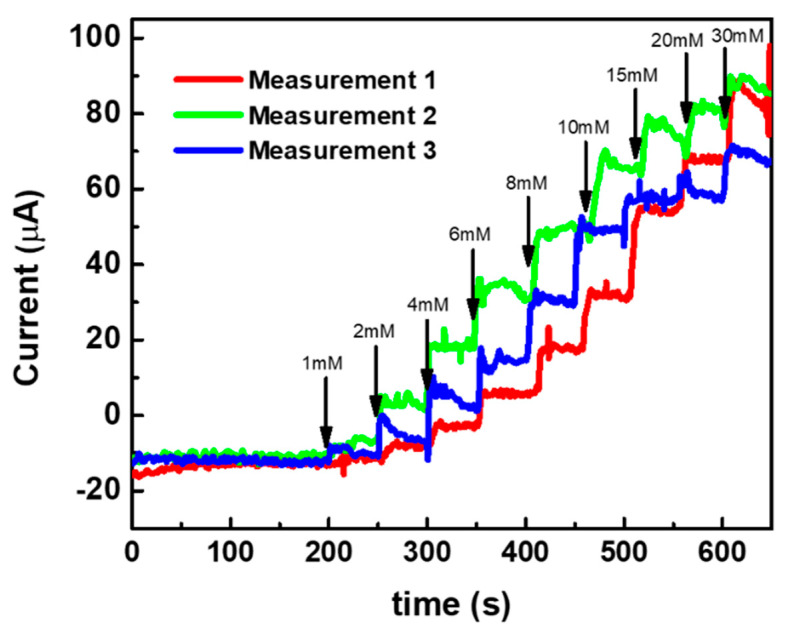
Response study of the sensor in ISF in a range of glucose concentration from 1 to 30 mM.

**Table 1 sensors-22-02787-t001:** Comparison of the electrochemical sensor performance characteristics recorded using the wireless device with the previous reports.

Electrode Configuration	Linear Range (mM)	Detection Limit (μM)	Sensitivity (μA mM^−1^ cm^−2^)	Reference
CuNPs/HD-CNT*f*	1 × 10^−5^–6.0	2.8 × 10^−5^	1.942	[29]
PANI@CuNi/GC	0.1–5.6	0.2	1030	[30]
CuO/PANI-NF/FTO	0.25–4.6	N.R	N.R	[31]
Au/GO	0.1–2, 2–16	0.025	5.20, 4.56	[32]
NiO-NPs@FTO	0.1–1.2	1.0	0.039	[33]
Cu-MOF	6 × 10^−5^–5.0	0.01	89.0	[34]
Co/MOF/NF	1 × 10^−3^–3.0	1.3	10.886	[35]
ZIF-N_2_	0.1–1.1	5.69	227	[36]
Ag0.5%@ZIF67%/GC	1 × 10^−3^–1.0	0.6	379	[37]
Au/Pt-black/Nf	1.0–30.0	480	145.33	This work
(*Bare AuMN as CE*)				
Au/Pt-black/Nf	1.0–10.0	268	445.75;	This work
(*Pt-black coated–AuMN as CE*)	15–30.0		165.83	

## Data Availability

Not applicable.

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
