# Peer review of "Sensitive Electrochemical Non-Enzymatic Detection of Glucose Based on Wireless Data Transmission"

_sensors, 2022, doi:10.3390/s22072787_

Round 1

Reviewer 1 Report

Initially, I congratulate the authors for the manuscript presented.

I will make 2 considerations about the manuscript:
1. Application to real samples: I suggest that the authors apply to real samples.
2. Validation studies: I suggest authors to insert validation studies in the manuscript.

Based on these 2 aspects, I do not recommend the publication of this manuscript.

Reviewer 3 Report

After carefully reading the manuscript, I recommend its publication after some necessary revisions. Detailed comments are listed as follows:

(1) "Introduction" needs to be more complete with some investigations, such as relevant articles especially published for non-enzymatic glucose sensors. Improve the introduction section by including the further references.

(2) The color of scale bars in Figure 4 is similar to that in SEM images, which makes it difficult to see clearly. The authors should recalibrate it.

(3) The element symbols in Figure 5 are not clear. The authors should modify them.

(4) How is the repeatability of the glucose sensor? And the long-term storage stability of the sensor should be evaluated.

(5) The author should evaluate the reliability of the proposed enzyme-free glucose sensor in human blood serum for practical application.

Reviewer 4 Report

The manuscript describes the development of a non-enzymatic sensor based on microneedles modified with black-Pt and recovery with Nafion. The results show promising results. However, below are some suggestions that may be considered before publication:

  1. It is unclear how the sensor can selectively act in glucose determination. I suggest that the authors add more information about this, including possible mechanism.
  2. It would be important for the authors to add a figure with the glucose measurement with and without nafion, including in the selectivity test, to verify its efficiency in this parameter.
  3. I suggest a figure with electrochemical characterization of material in each step of construction
  4. The current response for 5 mM glucose in selectivity study is very higher than obtained in calibration curve. Do the authors have an explanation for this?
  5. I suggest authors add a recovery test preferably on real sample.
  6. I think it would be important for the authors to add data from other studies by this research group involving similar non-enzymatic sensors (references 5-7). And still in the introduction, the difference of the proposed sensor in relation to the others of this research group could be highlighted.

Round 2

Reviewer 1 Report

The authors answered all questions.

Reviewer 2 Report

The revised manuscript shows great improvement.

Reviewer 3 Report

I have no comment, and this revised manuscript can bes accepted.

Reviewer 4 Report

The authors did the corrections suggested